# Chytrid fungi distribution and co-occurrence with diatoms correlate with sea ice melt in the Arctic Ocean

Estelle S. Kilias [1,2 ✉], Leandro Junges[3,4], Luka Šupraha [5], Guy Leonard[2], Katja Metfies[6,7] & Thomas A. Richards[2]

Global warming is rapidly altering physicochemical attributes of Arctic waters. These changes are predicted to alter microbial networks, potentially perturbing wider community functions including parasite infections and saprotrophic recycling of biogeochemical compounds. Specifically, the interaction between autotrophic phytoplankton and heterotrophic fungi e.g. chytrids (fungi with swimming tails) requires further analysis. Here, we investigate the diversity and distribution patterns of fungi in relation to abiotic variables during one record sea ice minimum in 2012 and explore co-occurrence of chytrids with diatoms, key primary producers in these changing environments. We show that chytrid fungi are primarily encountered at sites influenced by sea ice melt. Furthermore, chytrid representation positively correlates with sea ice-associated diatoms such as *Fragilariopsis* or *Nitzschia*. Our findings identify a potential future scenario where chytrid representation within these communities increases as a consequence of ice retreat, further altering community structure through perturbation of parasitic or saprotrophic interaction networks.

[1] University of Exeter, Bioscience, Living System Institute, Exeter, UK. [2] University of Oxford, Department of Zoology, Oxford, UK. [3] Centre for Systems Modelling and Quantitative Biomedicine, University of Birmingham, Birmingham, UK. [4] Institute of Metabolism and Systems Research, University of Birmingham, Birmingham, UK. [5] University of Oslo, Department of Biosciences, Oslo, Norway. [6] Alfred Wegener Institute for Polar and Marine Research, Bremerhaven, Germany. [7] Helmholtz Institute for Functional Marine Biodiversity, Oldenburg, Germany. ✉email: E.kilias@exeter.ac.uk

The marine environment is a vast reservoir of microbial diversity[1,2] with the Arctic Ocean a hotspot of biodiversity[3]. Climate-driven changes to local physicochemical properties are profound in the Arctic Ocean, for example, leading to increases in temperature, light availability and decreases in sea ice concentration and surface water salinity. These changes have been shown to alter microbial community structure[4–6].

Among marine eukaryotic microbes, 'marine-derived fungi' (a term referring to the sampling provenance of these microbes) are under-studied and insights into their ecological role are lacking. Recently, marine-derived fungi have received an increase in attention after high-throughput diversity tag-sequencing has revealed an uncharacterised diversity of fungi and fungal-like ribosomal RNA (rDNA) phylotypes in marine environments[7]. The discovery of this diversity has led to the hypothesis that bona-fide marine fungi (fungi recovered from sea water, which represent a functional player in the marine ecosystem, rather than a spore in transit) operate as a trophic 'bridge' for carbon and nitrogen substrate transfer between recalcitrant phytoplankton such as diatoms and wider nodes on the marine food-web, e.g., zooplankton[8,9]. One paraphyletic group of fungi called informally the chytrids, which can form flagellated zoospores capable of chemo- and photo-tactic swimming behaviour[10,11], operates as prominent parasites, saprotrophs, or both (through necrotrophic infection of host) in aquatic systems[12,13]. In the following text, the term chytrids will be used as a shorthand exclusively for members of the Chytridiomycota group. Other zoosporic fungi include members of the group Blastocladiomycota or Crypto-mycota/Rozellomycota but these groups are not a major focus of this study.

Putative chytrid groups have been detected using both microscopy and environmental DNA sequencing in deep-sea sediment, water column and sea ice samples[7,14–17]. Chytrids have been reported to have an effect on phytoplankton population dynamics through infection, causing changes in the timing and duration of blooms[12,18]. Diatoms are known to form vast blooms, influencing environmental nutrient flux by exporting carbon to the deep-sea[19]. Changes in diatom bloom progression are therefore expected to have cascading implications for the marine ecosystem and associated biogeochemical cycles in many regions of the marine environment. Specifically, parasitism of diatom species by chytrids has been shown to alter diatom bloom dynamics in marine- and fresh- waters[12,13]. Documented diatom host genera for chytrids include, for example, *Chaetoceros sp.*, *Cylindrotheca sp.*, *Navicula sp.*, *Nitzschia sp.* and *Pseudo-nitzschia sp.*[20–23]. In situ experiments, addressing the effect of abiotic and biotic factors on susceptibility of diatom to chytrid infections

have shown an influence of light-stress and host densities[22,23], factors which can potentially be driven by ice melt but this relationship has not been demonstrated from wider environmental sampling in ice melt conditions. In the Arctic Ocean, many of the diatom genera (e.g., *Nitzschia*, *Pseudo-nitzschia*, *Chaetoceros*) shown to be susceptible to chytrid infections include major contributors to phytoplankton biomass[24].

In this study, we investigated chytrid-rDNA distribution patterns in the Arctic Ocean in relation to abiotic (e.g., temperature, sea ice and salinity) and biotic parameters (wider protist community structure and specifically diatom distribution). The strategy includes both, Deep Chlorophyll Max (DCM) and Under-Ice Water (UIW) sampling in order to study the co-occurrence of fungi and phytoplankton in waters influenced by ice melt and surface water freshening. We combine sampling of both the eukaryotic general V4 SSU rDNA and fungal-specific ITS2 region with high-throughput Illumina sequencing, in order to assess the diversity and distribution of fungal sequences and identify environmental parameters that putatively drive this component of the marine microbial community. As part of this work, we test the hypothesis that patterns of sequence variant representation in these groups relate to sea ice melt conditions, identifying a correlation between chytrid fungi and diatom community structure with sea ice melt conditions.

## Results and Discussion

**Oceanography and protist community composition.** In summer 2012, the Arctic Ocean experienced an unprecedented decline in sea ice, resulting in a record sea ice minimum[5]. It has previously been shown that profound sea ice melt in summer 2012 led to increases in the light conditions and nutrient supply, affecting the microbial community and increasing net primary productivity of these waters[5,25]. Representative samples from across different Arctic oceanographic regions and encompassing multiple depths were sampled over a time span of three months, allowing us to compare a range of: communities, sea ice melt conditions, and temporal samples. For description of the oceanographic context of the samples please see Fig. 1, Table 1, Supplementary Fig. 1 and the methods section of this paper.

To understand the wider community dynamic of the environments sampled, the microbial community structure was assessed using eukaryotic V4 SSU rDNA[26] targeted PCR followed by high-throughput Illumina sequencing. The non-dimensional metric scaling (NMDS) ordination analysis comparing the 'V4-identified' community structure displayed differences in community diversity in the DCM samples (area shaded pink—Fig. 2a) compared to the UIW samples (area shaded grey—Fig. 2a). However, all UIW

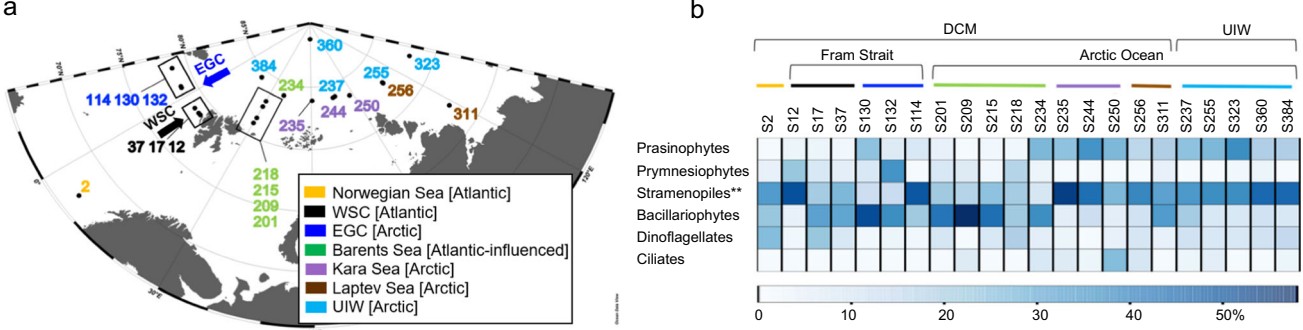

**Fig. 1 Characterisation of the sampling stations by V4 rDNA tag sequence diversity and abiotic characteristics. a** Sampling locations on the expedition ARK-XXVII, cruise leg 2 and 3 onboard the *RV Polarstern* in summer 2012; East Greenland Current (EGC); West Spitsbergen Current (WSC). **b** Major protist group distribution identified from V4 rDNA tag sequencing, shown as relative representation [%] at under-ice water (UIW) and deep chlorophyll max (DCM) sampling sites. **The 'stramenopiles' category shown excludes V4 sequences assigned as Bacillariophytes (Diatoms). See Supplementary Fig. 1 for more information on taxonomic composition of the V4 sequencing results.

**Table 1 Abiotic environmental properties (depth (D); temperature (T); salinity (S) of the 22 sampling stations sampled between June and September 2012.**

| Station (S) number | 2 | 12 | 17 | 37 | 114 | 130 | 132 | 201 | 209 | 215 | 218 | 234 | 235 | 244 | 250 | 256 | 311 | 237ᵃ | 255ᵃ | 323ᵃ | 360ᵃ | 384ᵃ |
|---|---|---|---|---|---|---|---|---|---|---|---|---|---|---|---|---|---|---|---|---|---|---|
| Date [dd/m] | 17/6 | 20/6 | 20/6 | 22/6 | 06/7 | 08/7 | 09/7 | 05/8 | 06/8 | 07/8 | 07/8 | 12/8 | 13/8 | 16/8 | 18/8 | 20/8 | 01/9 | 14/8 | 20/8 | 04/9 | 22/9 | 28/9 |
| D [m] | 15 | 30 | 32 | 10 | 30 | 20 | 12 | 30 | 32 | 17 | 50 | 22 | 26 | 30 | 30 | 20 | 20 | 0 | 0 | 0 | 0 | 0 |
| T [C] | 8.97 | 1.73 | 4.47 | 4.82 | −1.57 | −1.63 | −1.49 | 0.11 | −0.04 | −1.11 | −1.75 | −1.52 | −1.7 | −1.58 | −1.68 | −1.68 | −0.27 | −1.7 | −1.59 | −1.62 | −1.8 | −1.8 |
| S [PSU] | 35.09 | 34.71 | 35.13 | 35.11 | 31.55 | 32.89 | 32.48 | 34.25 | 34.27 | 33.26 | 34.26 | 34.04 | 34.20 | 34.17 | 34.13 | 33.74 | 32.1 | 32.9 | 32.61 | 30.54 | 33.07 | 32.83 |
| Ice [%] | 0 | 0 | 0 | 0 | 99 | 91 | 100 | 0 | 0 | 52 | 91 | 94 | 98 | 90 | 73 | 93 | 0 | 80 | 70 | 60 | 100 | 100 |

ᵃSampling stations were taken from under the sea ice (UIW) while all other stations were sampled from the deep chlorophyll max (DCM).

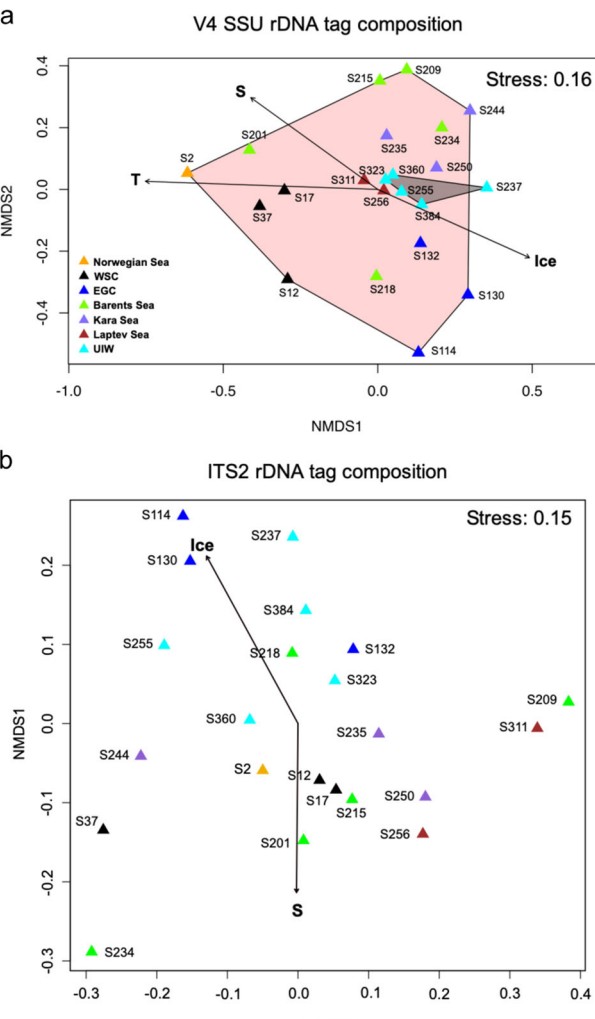

**Fig. 2 Community structure differences across oceanographic sampling stations.** Two-dimensional nMDS plot of **a** the microbial eukaryotic community structure (derived from V4 tag sequencing) calculated with the Bray Curtis dissimilarity index ($R = 0.16$). Temperature (T), salinity (S) and sea ice concentration (Ice) affected the community structure significantly ($p \le 0.001$), framed areas indicate sampling provenance within the DCM (pink) or UIW (grey) **b** the fungal community structure (derived from ITS2 tag sequencing) calculated with the Bray Curtis dissimilarity index ($R = 0.15$) and with salinity and sea ice concentration shown to be major factors affecting the community structure ($p \le 0.05$).

samples clustered within the DCM samples in the NMDS range, indicating no distinct community structure in UIW compared to DCM samples. Ice-free Atlantic water communities such as S2 (Norwegian Sea), S12–S37 (West Spitsbergen Current; WSC) and S201 (Barents Sea) grouped separately (on the '– side' of the x-axis of the NMDS plot—Fig. 2a) from communities of the East Greenland Current (EGC; S114–S132) and other Arctic regions (ANOSIM: $R = 0.67$, $p = 0.001$). Single oceanographic regions showed some structuring effect, with samples originating from the same region (indicated by identical colour code—Fig. 2a) grouping closer to each other (ANOSIM: $R = 0.35$, $p = 0.003$). MANTEL tests of distance matrices demonstrated a correlation between environmental parameters and 'V4-identified' community structure. A subsequent principal component analysis for the purpose of identifying environmental factors which showed influence on community structure identified that the linked abiotic characteristics of temperature, salinity and sea ice concentration are

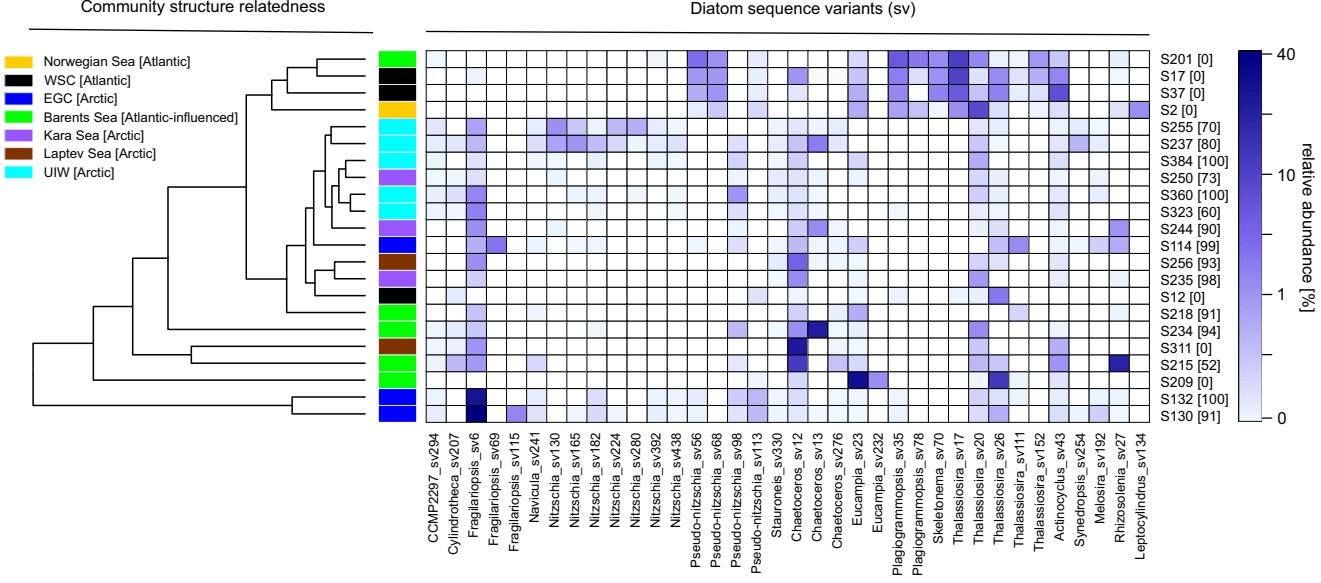

**Fig. 3 Heatmap showing the V4 read representation of the 25 most abundant diatoms.** The diatoms were detected in the deep chlorophyll max in different oceanographic regions (orange, black, blue, green, purple, brown) and under sea ice (light blue), see the colour coded key. Community structure related cluster dendrogram was calculated using the Euclidean distance and, therefore, reflecting the diatom community composition similarities/differences of the sampling stations. % sea ice cover at the sampling station is listed in square brackets besides the sampling station name, down the y-axis.

significant factors ($p \leq 0.005$) for driving the V4 eukaryotic community structure, consistent with other studies[27–31].

The major eukaryotic phytoplankton groups were distributed as show in Fig. 1b and Supplementary Fig. 1, with prasinophytes (Chlorophyta) recovered at a higher relative representation in Arctic waters of the EGC (S114–S132) and a subset of Arctic sampling stations (specifically at S234–S384). Bacillariophytes (diatoms) showed a higher relative representation in the Fram Strait (EGC, e.g., S114–S130 and WSC, e.g., S17 & S37) and the Barents Sea (S201–S234). Other stramenopiles, mainly of MAST affiliation (e.g., MAST-6; Supplementary Fig. 1) showed increased relative representation in the Eastern Arctic Ocean waters (S235–S384). Specifically, within the DCM samples, the phytoplankton community structure also showed a strong differentiation in cell size between nanoplankton (2–20 µm, e.g., prymnesiophytes mainly *Phaeocystis pouchetii*) dominated Atlantic/Atlantic-influenced waters (WSC, Barents Sea, e.g., S12–S218) and picoplankton (0.2–2 µm, e.g., prasinophytes mainly *Micromonas pusilla*) dominated Arctic waters (EGC and from S234–S384). The distribution pattern is consistent with previous work showing a correlation with nutrient and light-limited conditions, which have been shown to drive similar patterns of protist biogeography linked with selection for 'frugal' picoplankton species, specifically *M. pusilla*[6,28–30].

A similar pattern, was also recovered within the micro-phytoplankton (20–200 µm, e.g., bacillariophytes) distribution, where diatom diversity was demarcated between these regions where specific *Thalassiosira* sequence variants (centric diatom) dominate the diatom communities in the Atlantic waters (e.g., Norwegian Sea: S2 and WSC: S17–S37—Fig. 3), and specific *Fragilariopsis* (pennate) & *Chaetoceros* (centric) sequence variants dominate diatom communities in Arctic waters (e.g., EGC: S114–S132, Kara Sea: S235 and S244, Laptev Sea: S256 and S311, and UIW—Fig. 3). *Fragilariopsis*, is widely distributed in cold environments and often associated with sea ice[32]. Consistent with this observation, sampling sites that recovered high-relative representation of *Fragilariopsis* in the V4 rDNA diversity tag profiles were characterised by sea ice and low salinity (e.g., EGC:

S132, S130 and UIW: S237, S255, S323, S360—see Fig. 3), suggesting sea ice melt conditions.

Samples with high sea ice concentrations (see Figs. 1c and 3) were also commonly characterised by the presence of genera of other, known, sea ice-associated diatoms such as *Nitzschia* sp. and *Navicula* sp. Sea ice algae are important primary producers in the Arctic Ocean, capable of growth in low-light conditions and living in brine channels and ice pores[24]. One characteristic attribute of sea ice algae is the production of extracellular polymeric substances, heterogeneous mixtures of biopolymers (e.g., polysaccharides, proteins, nucleic acids and phospholipids), which are exuded in higher concentrations during periods of low-nutrient availability and which allow for cryoprotection and potential floatation under sea ice based on $O_2$ inclusion[33–35]. At UIW sampling sites, two pennate sea ice genera dominated the V4 rDNA sequence tag profiles: *Nitzschia* (e.g., S237 and S255) and *Fragilariopsis* (e.g., S323 and S360) along with *Chaetoceros*, *Navicula*, and the sub-ice genus *Melosira* to a lesser extent (Fig. 3). As these genera have been associated with sea ice[36], it is plausible that they have been released from the ice during melt conditions. In support of this scenario, *Nitzschia* sp. was also recovered in the sympagic community sampled close to our S237 and S255 samples by Stecher et al.[37]. Taken together, our sample strategy covers a wide geographic region of Arctic waters and demonstrates that the samples recovered encompass contrasting environmental conditions, including environments subject to sea ice melt changes and associated changes in the primary producer communities, including exposure to sea ice-derived diatoms.

**Fungal community structure based on rDNA tag sequencing.** To understand fungal community structure, we first conducted fungal-specific PCR of the ITS2 region of the rDNA gene cluster followed by Illumina amplicon sequencing for all our DNA samples. To summarise these findings, we investigated taxonomic assignment of the ITS2 sequences sampled, focusing on putative fungal sequence variants with higher representation (i.e., sequence variants with ≥1% of all sequences reads recovered—Fig. 4).

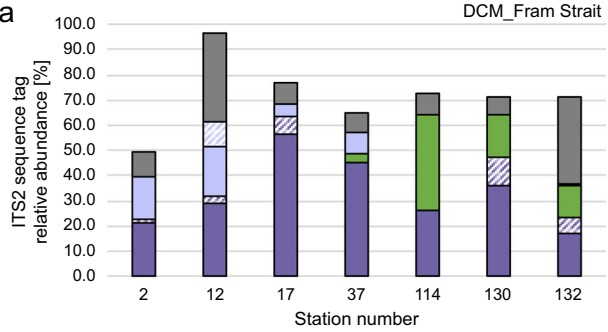

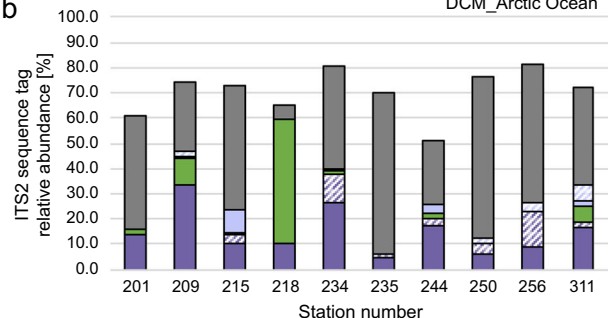

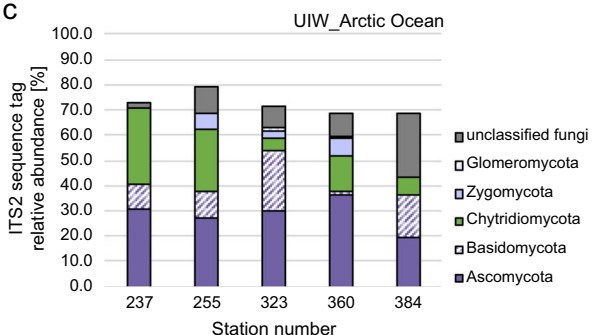

**Fig. 4 Distribution of fungi (≥1% abundance) based on the ITS2 marker gene sequencing identifying putative patterns of fungal diversity across the sampling stations. a**, **b** Relative representation of fungal groups in the deep chlorophyll max (DCM) of the Fram Strait, including the West Spitsbergen Current (WSC) and East Greenland Current (EGC) (**a**) and the Arctic (**b**). Please note that S218 is an outlier in the Artic samples and contains a large relative representation of chytrid sequences. **c** Relative representation of fungal groups in the under-ice samples (UIW).

Our fungal-specific ITS2 analyses recovered a large diversity of sequences identified as 'unclassified fungi', suggesting the marine environments sampled harbour a diversity of previously unsampled evolutionary distinct fungal taxa or eukaryotic microbes that share similar ITS2 sequence characteristics to fungi (Fig. 4). The highest proportions of unclassified fungi were detected in the Arctic DCM samples (Ø42.2%) and lowest in UIW (Ø13%) samples. Current ITS2 database information prevents us from further investigation of the nature of this diversity or appropriate tests to confirm these sequences are correctly annotated as 'fungal'. High numbers of unclassified fungi in Arctic DCM and UIW samples imply that the diversity of marine-derived fungi is still under-explored, particularly in remote areas such as the Arctic Ocean. This is consistent with other studies that have revealed a diversity of uncharacterised fungi and fungal-like sequence in marine environments (see ref. [7]).

Glomeromycota and Zygomycota were not-detected at most stations and if present represented a low-average relative sequence representation within our samples of 1.2 and 3.8%, respectively. Basidiomycota showed a general low presentation in DCM samples, not exceeding 7% (Fig. 4a, b: EGC: 6.6%, WSC: 3.3% and Arctic: 3.9%). Higher representation of basidiomycetes was identified in the UIW samples (Ø12.6%), suggesting that basidiomycetes are associated with sea ice melt (Fig. 4c).

Ascomycota displayed a stable distribution of ~29% in UIW samples (Fig. 4c). Representation of the Ascomycota in the DCM sample varied considerably with average relative representation in the Fram Strait of 26.4% (EGC; S114–S132) and 43.5% (WSC; S12–37) and 14.7% in the Arctic (S201–S311). Previous studies of the Arctic Ocean reported ascomycetes on drift wood, (e.g., Leotiomycetes and Sordariomycetes) and in subglacial ice (Saccharomycetes)[38]. All three groups were recovered within the ITS2 data set. One plausible explanation for the high distribution in the WSC is the geographic location and proximity to Svalbard, characterised by coastal influence such as river runoffs and debris transport of plant matter.

The Chytridiomycota (Fig. 4) were detected with high-relative representation in many of our data sets and this group was dominated by sequence variants assigned to the class chytridiomycetes. Out of 127 Chytridiomycota sequence variants (sv) identified, 86 (72%) were classified to the orders Rhizophlyctidales (68–54%) and Chytridiales (23–18%) based on DADA2 taxonomic assignment. Follow up phylogenetic analyses of the ITS2 data are presented in Supplementary Fig. 2, showing that these two groups branch paraphyletically to each other but branch with known taxa of the class Chytridiomycetes (e.g., *Batrachochytrium*). Given the high variability of the ITS2 marker and short sequence alignment, we suggest these phylogenies should be interpreted as tentative. Average representation of Chytridiomycota differed strongly in the Fram Strait (Fig. 4a), with 22.5% in the EGC (S114–S132) and only 1.2% in the WSC (S12–S37). In the Arctic, Chytridiomycota accounted for ~8.3% (DCM; S201–S311), but were predominately recovered at a single sampling site (S218) and this group was consistently recovered at high-relative representation in the UIW samples (Fig. 4c) with Ø16.2% among the fungal community detected, suggesting a co-association with ice and sea ice melt conditions.

**Phylogenetic placement of marine-derived chytrid V4 sequences.** The V4 sequence diversity analysis consistently recovered a wide diversity of eukaryotic groups (see Supplementary Fig. 1). A relatively small proportion of these were identified as Fungi (Supplementary Fig. 3). As our ITS data indicated a relationship between chytrid detection and/or relative abundance and ice and ice melt conditions we used our V4 SSU rDNA tag sequencing data to further explore the chytrids detected. Overall, fungal group distribution supported our previous findings from the ITS2 analysis, with chytrids showing higher relative representation/abundance at low-salinity sampling sites of the EGC (e.g., S114–S132—Supplementary Fig. 3) and in UIW (e.g., S237–S384—Supplementary Fig. 3).

The advantages of V4 tags over ITS sampling is that it allows for improved phylogenetic analysis for the purpose of taxonomic comparisons. This is because SSU rDNA sequences have improved database sampling of taxonomically identified fungi and this gene region allows for improved phylogenetic resolution in comparison to trees generated from ITS sequences[39]. A total of 25 different sequence variants, five of which were recovered with a representation of ≥0.1% (sv261, sv479, sv671, sv714 and sv764) were sampled among the wider V4 defined eukaryotic community and were assigned to the class Chytridiomycetes. These sequence variants were placed in a Maximum Likelihood phylogenetic tree (colour coded in green) with published chytrid sequences (Fig. 5)

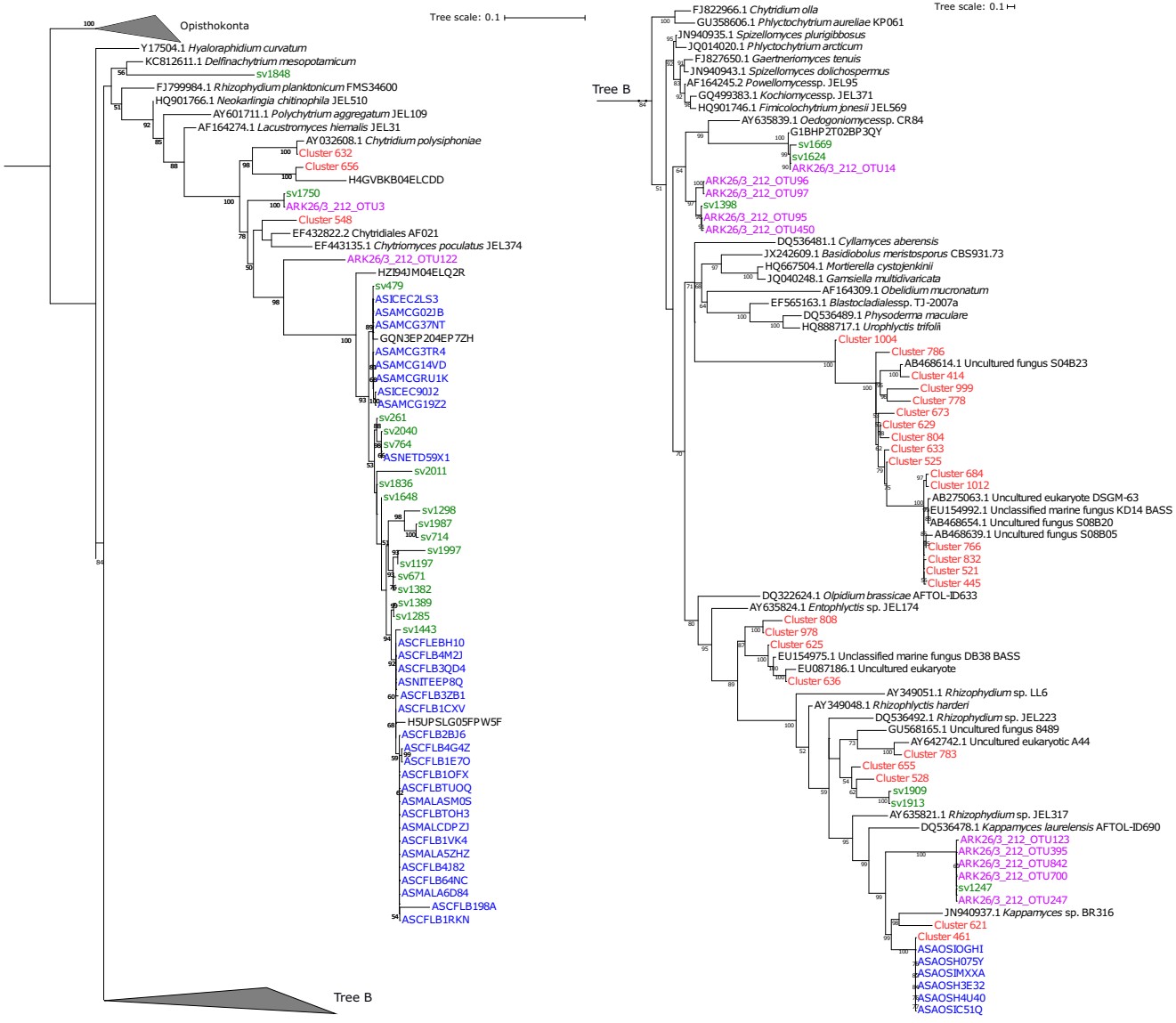

**Fig. 5 Phylogenetic classification of chytrid sequence variants detected in the V4 sequencing data set.** The phylogenetic tree was computed based on the maximum likelihood method in IQ-Tree including 1000 bootstrap replicates from a masked alignment of 154 taxa and 399 characters. Chytrid-like sequence variants identified as part of this study are highlighted in green (see phylotypes labelled sv#). Chytrid sequences from the BioMarKs study[14] are coloured in red (see phylotypes labelled Cluster#) , from meltpond samples[40] in purple (see phylotypes labelled ARK#), and from Comeau et al. in blue (see phylotypes labelled AS#) [14].

derived from BioMarKs; European sampling locations (red)[17], Arctic sampling locations (blue)[14], and Arctic meltpond aggregates (purple)[40]. The majority of the chytrid sequence variants (17), including the five most highly represented sequences, formed one major branch together with OTU sequences from the western Arctic (100% BS support)[14]. Furthermore, additional chytrid-like sequence variants recovered in this study branched with *Rhizophydium/Kappamyces* sequences (sv1909, sv1913 and sv1247), which includes a number of sequences previously obtained from meltpond aggregates, which were sampled in the previous year on a similar sampling expedition (ARK26/3)[40]. Only a few chytrid sequences from European sampling campaigns (e.g., BioMarKs' Cluster 461, 528, 655) branched in proximity to sequences from this or the Arctic Comeau study, with all of them originating from the Norway BioMarKs sampling site. No distinct diversity patterns were observed between DCM or UIW samples.

Based on individual phylogenetic analysis (Fig. 5) it was not possible to classify if any of the ITS2 sequences recovered could be paired with the V4 sequences sampled. To try and pair the different sections of the rDNA gene cluster sampled, we used local BLASTn searches with the chytrid V4 and ITS2 sequences from this paper as search seeds against the NCBI 'nt' and 'env_nt' and the 243 Tara Ocean Meta genome Assemblies (https://www.ebi.ac. uk/ena/about/tara-oceans-assemblies). Default settings were used, with tabulated output and the e-value gathering threshold was set at 1e-10. The BLASTn results were then filtered using the tool 'awk' to show any results with ≥2000bp hit length and 90% ID in an attempt to find DNA contigs that would allow us to infer a join between the chytrid V4 and ITS2 sequences. This approach returned no hits. To further explore putative ITS2-V4 associations we conducted a separate phylogenetic analysis for both the ITS2 and the V4 markers including ribosomal gene clusters sampled from a range of fungal genome projects of taxa related to the

chytrids (Supplementary Figs. 2 and 4). Again, this analysis failed to identify phylogenetic association that would allow us to infer that the ITS2 and V4 were from the same lineage, although it consistently confirmed that both the ITS2 and V4 chytrid sequences sampled, largely from under-ice water samples and the freshening East Greenland Currents, were phylogenetically associated with the chytridiomycetes.

**Drivers of Arctic fungal molecular diversity and provenance**. The ITS2 fungal community composition varied across sampling sites, with salinity and sea ice concentration identified as significant environmental drivers of community difference ($p \leq$ 0.05—Figs. 2b and 4). In particular, ITS2 defined fungal communities from the EGC (DCM; e.g., S114, S130) and UIW (e.g., S237, S255 S384) differed from communities from the WSC (DCM, S12–S37) and Arctic (DCM; e.g., S201, S234, S256) sites. This demonstrates that fungal sequence variant distribution correlated with fresher-waters (Fig. 2b), while no significant difference in community composition relative to geographic origin was observed (Fig. 2b; ANOSIM: $R = 0.07$, $p = 0.27$). These findings suggest that sea ice-associated or sea ice melt influenced waters display an altered ITS2 defined fungal community.

Chytrids, including orders like Chytridiales, Rhizophydiales, Mesochytriales, Gromochytriales and Lobulomycetales, have been found in different Arctic habitats including: open water sites, marine sediments, sea ice and wood substrates[38,41–43]. The distribution of chytrids in the present study was patchy across sampling sites, consistent with the findings of Comeau et al.[14] for the Western Arctic. Distribution patterns suggest an influence of sampling provenance with chytrids, on average, more highly represented at UIW sampling stations (Fig. 4c). Within the DCM, chytrids were mainly found in the EGC (Fig. 4a; S114–S132). As discussed above, one commonality of all sampling sites with high chytrid relative representation is the comparatively low salinity of these waters (see Table 1). This result is consistent with data from Burgeaud et al.[44] who have demonstrated a higher fungal diversity and relative representation at lower salinity sampling sites across a salinity gradient in the Delaware bay. These data are therefore consistent with the proposition that salinity appears to have an important impact on chytrid diversity and community representation[45].

Other abiotic parameters reported to influence chytrid distribution are light, temperature and/or nutrients[16,22] and indeed model flagellated fungi have been shown to possess a sophisticated set of chemotactic and photo-tactic responses allowing them to navigate in response to light and nutrient gradients (e.g., refs. [10,11]). In this study, neither light availability (identified using sea ice concentration as a proxy) nor temperature showed a strong trend for chytrid distribution.

Consistent with the ITS data, the V4 SSU rDNA phylogenies identified a proportion of unclassified chytrid-like sequences (Fig. 5). The majority of the 'V4-chytrids' identified here (green) clustered with OTUs from the Western Arctic Ocean (blue)[14] and from those originating from an Arctic meltpond aggregate (purple)[40]. These results demonstrate that the putative chytrids detected in this study grouped closer to SSU rDNA sequences sampled from cold waters, providing tentative evidence of a set of 'cold-water chytrids'. Furthermore, marine-derived fungi retrieved from the Western Arctic, displayed little overlap with representatives from Atlantic sites, suggesting that further biogeographic or environment-specific distribution patterns are evident[14].

Interestingly, basidiomycetes displayed a comparably stable distribution in UIW (Fig. 4c). In the marine environment,

basidiomycetes have been recovered in deep-sea sediments[46] and therefore are thought to be associated with physical substrates. One hypothesis for the higher relative representation of basidiomycetes at UIW samples could be that sea ice serves as physical substrate, releasing basidiomycetes, or substrates that basidiomycetes can colonise, during sea ice melt. This explanation is consistent with a study conducted at the North Pole, where a high diversity of basidiomycetes were found in sea ice[47]. Floating algal aggregates under the sea ice can further form such substrates. As mentioned previously, high production of exopolymers act to increase the 'stickiness' of algae, which makes them inherently prone to aggregation and sedimentation after floating under the ice[48,49], suggesting that sea ice algal aggregation may provide a substrate for basidiomycetes and other fungi in UIW environments—allowing for an osmotrophic/lysotrophic trophic interaction[50], which otherwise would not be viable on smaller and/or dissolved particles. Consistent with this observation, basidiomycetes in this study demonstrate low-relative representation at the single multi-year ice (MYI) station S360 (Fig. 4c), where elongated filaments of *Melosira arctica* rather than free-floating aggregates were found[51].

**Co-occurrence of diatom and chytrid sequence variants**. Previous work has demonstrated a co-association between chytrid taxa and diatoms in the Humboldt Current[12]. In the Arctic, the diatom V4 diversity profile in the environments sampled seemed to be influenced by a range of factors, including local sea ice concentration, sampling depth/strategy (DCM vs. UIW) and/or region. Highest diatom representation in the V4 diversity profile (>40%) were observed at ice-free stations (Fig. 3; e.g., S17, S37, S201). The centric diatom *Thalassiosira*-dominated diatom sequence assemblages at sampling stations influenced by Atlantic waters (e.g., Norwegian Sea (S2), WSC (S12–S37) and ice-free Barents Sea (S201 and S209), while the sea ice-associated diatom *Fragilariopsis* was most highly represented (>30%) in the EGC (e.g., S130, S132) environment, influenced by Arctic waters, higher sea ice concentrations and lower salinities, possibly as a product of sea ice melt. In comparison with the DCM samples, UIW samples showed a reduced relative representation of diatoms in the V4 diversity profiles, largely composed of 'sea ice diatoms' such as *Navicula* sp., *Nitzschia* sp., *Chaetoceros* sp., and *Fragilariopsis* sp. Out of the five UIW stations, S384 was sampled late in the season, and as such not sampled during sea ice melt but during sea ice formation. As a consequence, the diatom assemblage showed very few characteristic sea ice diatoms (<1%).

Correlation analysis to investigate chytrid distribution in association with potential hosts such as diatoms, chrysophytes or dinoflagellates was computed. This analysis included all sampling sites and putative host sequence variants showing a representation of at least 1% at one or more sampling site (Fig. 6). Considering that chytrid sequence variants in the V4 data set showed low-relative representation among the wider complex community, ranging between 0.01 and 0.7%, these results should be interpreted tentatively, providing only indications of putative co-occurrence. Interestingly, potential positive correlations between chytrids and diatoms were predominantly with pennate diatom species (Fig. 6a). Among pennate diatoms, *Fragilariopsis* (e.g., Fragilariopsis_sv6) showed the strongest potential positive correlations with chytrid sequence variants showing the highest relative representation of sequence reads (e.g., Chytridium_sv261 and Chytridium_sv764). Further positive correlations were recovered with *Chaetoceros*, *Nitzschia* and *Eucampia*; diatoms often associated with sea ice, while *Thalassiosira* (e.g., Thalassiosira_sv17 or Thalassiosira_sv111) gave rise to predominantly negative correlations. Positive

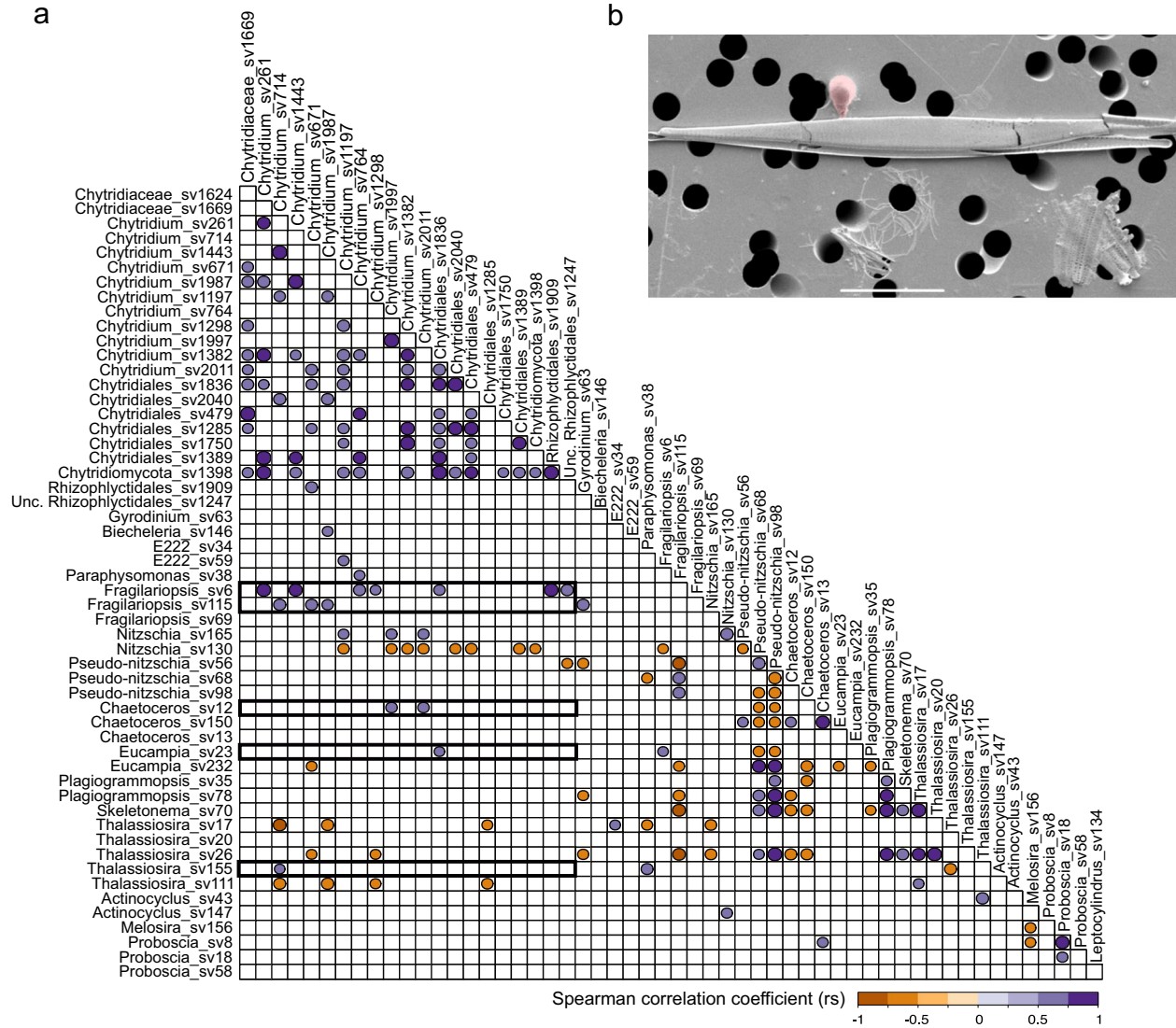

**Fig. 6 Evidence of host-chytrid co-association by correlation in the arctic samples. a** Spearman correlation coefficient (rs) plot of abundant chytrid, diatoms, chrysophytes and dinoflagellates sequence variants based on all DCM and UIW stations. A significance threshold of $p \leq 0.01$ was applied. Positive correlation coefficents between diatoms and chytrids are highlighted by a box. **b** Image shows pennate diatom with associated putative chytrid [zoo-] sporangium (false-colour red) sampled alongside our DNA sampling effort. This SEM was generated from a sample recovered in parallel to DNA recovered from central artic sample 491 using the same cell filtering process used for DNA extraction, hence the non-optimal SEM imaging.

correlation of three chytrid sequence variants were also found with one dinoflagellate and two chrysophytes, however their relative representation did not exceed 0.1%.

To further support our hypothesis that chytrid fungi are forming interactions with pennate diatoms under melting conditions, we searched associated samples that were collected in parallel to the DNA samples using the same filter-based approach used for DNA sampling for the purposes of 'crude' SEM analysis. Our analysis of these samples identified one candidate image suggestive of an interaction between a chytrid [zoo-] sporangium and a pennate diatom (Fig. 6b). To further explore this possibility of such interactions, we searched a collection of SEM images from meltponds aggregates from the AeN2018707 cruise on board RV Kronprins Haakon in summer 2018. These data show a series of images where a putative fungal-chytrid hyphal or rhizoid structure is shown penetrating a pennate diatom often as part of an aggregate and with the rhizoid/hyphae attached to a putative [zoo-]sporangium consistent with the

presence of such interactions in meltpond aggregate samples (Fig. 7a–f).

In order to further evaluate the role of diatoms as hosts for chytrids we made use of the 'Ocean Sampling Day 2014' data set, including 145 surface water sampling sites covering different locations and environmental scenarios. Computation of potential correlations between chytrids and the wider protist community showed a number of associations (Fig. 8). Out of 20 sequence variants affiliating to chytridiomycetes, three showed multiple significant correlations (Chytridiomycetes_sv1150, Chytridiomycetes_sv622 and Rhizophydiales_sv1879; $p = 0.05$) with different protist community members. A phylogenetic placement of these three sequence variants revealed a close phylogenetic relationship of one sequence variant (OSD_S1879; Rhizophydiales) with Arctic chytrids from the Comeau study[11] (Supplementary Fig. 2). Interestingly, out of 57 correlations in total, diatom co-occurrences account for ~40% of them, supporting the role of diatoms as dominant hosts with *Guinardia* (centric), showing the

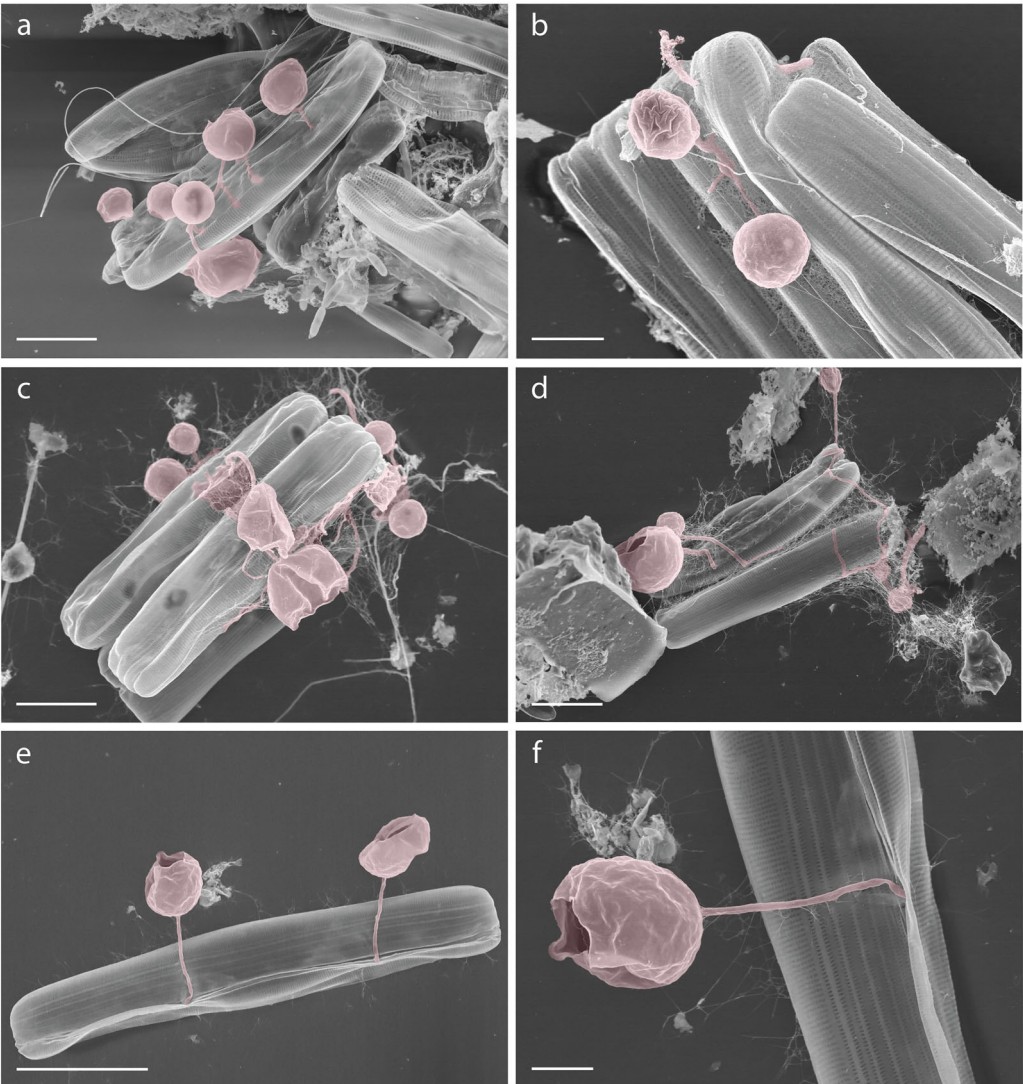

**Fig. 7 Scanning electron microscopy (SEM) images of pennate diatoms from meltpond aggregates putatively infected with chytrid-like fungal pathogens.** (**a**–**f**) false-colour red shows chytrid-like [zoo-]sporangium structures. **f** is a sub-section of **e**. Scale bars: **a**, **c**, **d** = 5 μm; **b**, **f** = 2 μm; **e** = 10 μm.

highest correlation. This global wide sample correlation analyses further confirm the chytrid community correlation in terms of co-occurrence with diatoms.

Chytrids have been shown to exert a top-down control on diatom community dynamics[8]. Considering, the importance of diatoms for the food-web, primary productivity and carbon export it is important to investigate the environmental factors affecting chytrid distribution and representation. In Hawaiian waters, eutrophic and mesotrophic conditions were found to correlate with chytrid infection of *Cyclotella* and *Chaetoceros*, while higher rates of attached putative chytrid zoosporangia were observed under *Thalassiosira*/*Skeletonema* predominance during austral spring in the Southern hemisphere (Humboldt Current)[12,52]. In this study, chytrids were primarily found at UIW sampling sites or EGC stations characterised by low salinity and a high prevalence of *Chaetoceros*, *Fragilariopsis* and *Nitzschia* spp., while low-relative representation of chytrid sequence variants were observed in *Thalassiosira*-dominated, ice-free water sites. This suggests that either abiotic conditions are directly driving the chytrid community composition or indirectly through biotic interactions, for example the composition of the diatom community, which is potentially shaping chytrid community dynamics through heterotrophic interactions.

In summer, the melting of Arctic sea ice leads to surface water stratification, salinity decrease and the formation of meltponds. In the Arctic Ocean, the distribution and population density of diatom species depends largely on the extent of light transmission through sea ice, the temperature, salinity, nutrient status, as well as the presence of heterotrophic protists and grazers. Changes in these environmental factors are hypothesised to trigger stress responses in diatoms and other primary producers, which in turn can putatively affect the susceptibility of these phytoplankton organisms to parasites. In situ infection experiments of chytrid and diatom species under various stress conditions demonstrate a higher infection density under light-stress conditions and a positive infection prevalence for *Chaetoceros* at low temperature (5 °C)[21,23]. In marine high-latitude environments, phytoplankton from north-west Iceland were prone to increased viral infection, predation and fungal parasitism under higher light and higher water temperatures[22], this is consistent with sea ice melt conditions. In the Western Arctic, a link between chytrid occurrence and algal stress in relation to increased light availability was found, particularly for pennate diatoms like

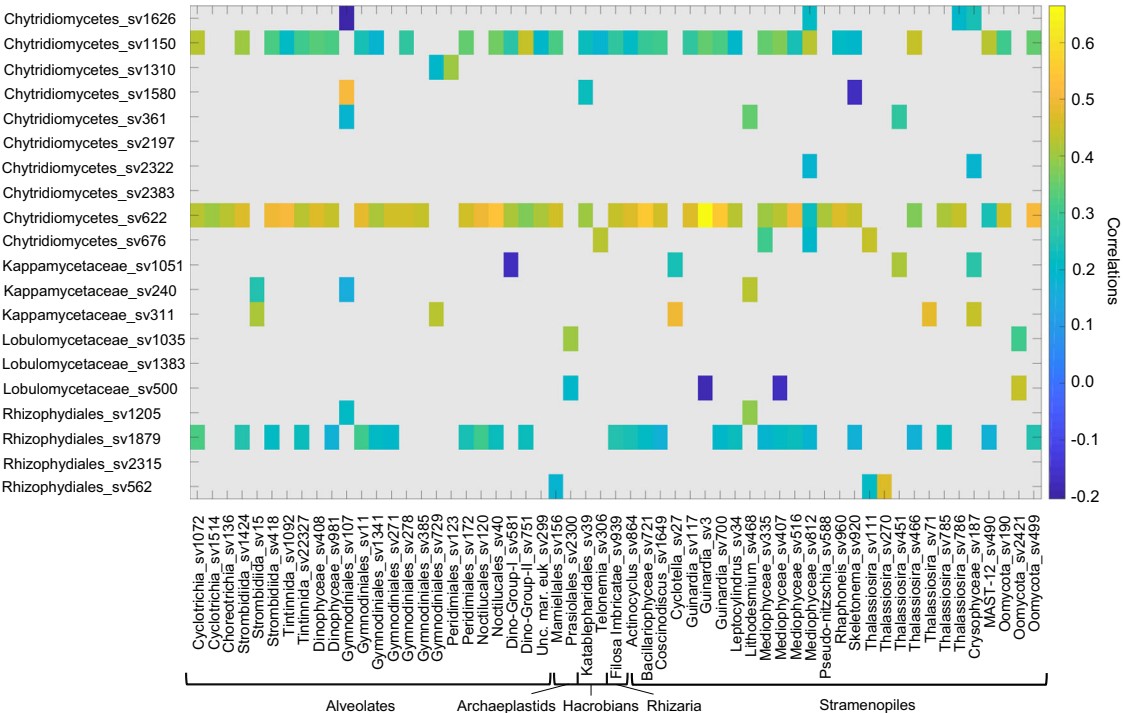

**Fig. 8 Correlation analysis of chytrid and protist sequence variants (sv) co-associations from the Ocean Sampling Day initiative in 2014.** Sequence variants have been filtered based on a minimum sequence number of 100 across all sampling sites, while a correlation coefficient threshold of 0.3 and a pseudo *p*-value of *p* = 0.05 have been adopted (the stramenopile group listed here contains include predominately diatoms).

*Pleurosigma* and *Navicula*[16,41]. These findings imply a potential increase in parasite infection under prolonged light exposure during Arctic summer months. The detection of chytrids in Arctic sea ice[16], therefore, suggests that occurrence of chytrids and specific diatom species is linked, either directly due a trophic relationship or indirectly derived by similar patterns of environmental selection. If it is the former and these patterns are driven by a trophic relationship, these findings collectively suggest that stress of host species induced by changes in environmental factors is a modifier of these interactions.

At UIW stations the highest representation of chytrids was associated with the highest representation of diatoms, including pennate sea ice algae and advanced sea ice melt. This pattern suggests that sea ice melt is releasing a diversity and abundance of stressed diatoms often in aggregate form, which are available for heterotrophic utilisation. Indeed, pennate diatoms of the genera *Nitzschia*, *Navicula* or *Fragilariopsis* were frequently observed in aggregates from first year sea ice melts from similar samples[51,53]. We therefore hypothesise that ice algae aggregates present an attractive micro-habitat for chytrids as shown in Fig. 7. The concentration of organic material and diatom cells is likely to trigger the recognition of chytrid zoospores and provides a habitat, which benefits osmotrophic/lysotrophic feeding[50] including parasitism. Supportive evidence for our hypothesis is the detection of chytrids in meltpond aggregates from 2011[40] (again see Fig. 7 for evidence of chytrid-diatom interactions in meltpond aggregates), which cluster close to chytrid sequence variants identified here (see Fig. 5 for phylogenetic data).

In summary, the objective of this study was to investigate distribution patterns of fungi, specifically chytrids in Arctic waters. These phylotypes were often found in association with ice melt conditions and in co-occurrence with specific diatom genera, suggesting a potential parasite-host relationship and/or evidence of saprotrophic interaction. The study also investigates these factors in relation to abiotic parameters. Our sampling makes use

of sampling during a record Arctic sea ice minimum in 2012. Salinity showed a strong influence on chytrid distribution, it is, therefore, reasonable to suggest that the interplay of salinity decrease and light increase during advanced sea ice melt has either directly (abiotic) or indirectly (biotic) influenced chytrid distribution and relative abundance. As such we hypothesise, free-floating algal aggregates under the sea ice, consisting of pennate diatoms forming putative micro-habitats for chytrid trophic associations. Furthermore, we observed a close phylogenetic relationship between chytrid sequence variants from the Arctic Ocean, identifying potential cold-water chytrid phylogroups. In a wider context, these findings suggest that chytrid prevalence in Arctic waters are likely to be altered under continuous sea ice melt conditions and freshening associated with climate change, leading to increased chytrid representation. In order to elucidate the role of chytrids in the Arctic Ocean under continuous sea ice melt, additional environmental studies are needed, including wider seasonal sampling (e.g., spring blooms) and different sea ice melt gradients and the sampling of algal aggregates associated with sea ice melt. Future work must broaden our understanding of potential host organisms, including other algae, e.g., chrysophytes or mixo-/heterotrophs like dinoflagellates and key groups of grazers.

## Methods

**Sampling of environmental samples.** Sea water samples were collected in the sea ice minimum year 2012 on board *RV Polarstern* on two cruise legs (ARK-XXVII/2 and 3). The first cruise leg, ARK-XXVII/2 includes one station in the Norwegian Sea and a transect through Fram Strait from 11°58.362′ to 11°5.09′ E longitude at ~78°50′ N latitude. The second cruise leg, ARK-XXVII/3 covered the Eastern Arctic Ocean, including sampling in the Barents Sea, and at the edge of the Kara Sea and Laptev Sea and at one high-Arctic sampling station (Fig. 1a). Throughout the manuscript, sampling stations are colour coded based on the oceanographic region. A total of 22 samples, including 17 DCM and 5 UIW samples, were taken between mid-June and the end of September 2012. The chlorophyll maximum layer was sampled with a CTD rosette (with conductivity, temperature and depth profiler parameters recorded) equipped with an array of 10 L Niskin bottles, while the UIW

was sampled with a portable CTD. From each depth, subsamples of 2 L were sequential filtered under 200 mbar onto 0.4 μm polycarbonate filters (Whatman) and stored at −80 °C.

**Oceanographic context of the samples**. Twenty-two water samples, including 17 DCM and 5 UIW samples, were collected from high-latitude oceanographic regions in Fram Strait and the Central Arctic Ocean. The study area was characterised by a high degree of variability in sea ice coverage, including large areas of sea ice melt (Fig. 1a, Table 1). Fram Strait is characterised by a bidirectional current system, the West Spitsbergen Current (WSC), which is mainly ice-free year-round, and the East Greenland Current (EGC), which is an area characterised by melting sea ice. Three DCM samples (S12, S17 and S37) were taken in the WSC, transporting Atlantic water northwards and three samples (S114, S130 and S132) in the EGC, transporting Arctic water and sea ice southwards. EGC stations showed between 91 and 100% sea ice coverage, an average temperature of −1.56 °C and salinity of 32.3 PSU (Table 1). No sea ice was observed at the WSC stations or at the sampling site in the Norwegian Sea (S2), located in Atlantic water. Here, the average water temperature and salinity was 3.6 °C and 35 PSU, respectively.

Except for station S360, all sampling stations in the Arctic Ocean were located in the marginal areas of the Barents Sea, Kara Sea and Laptev Sea in Nansen Basin (Fig. 1a). Temperature at Arctic DCM sampling stations ranged between −1.75 (S218) and 0.11 °C (S201), while salinity was stable with ~34 PSU at all stations except for S311, which had a 32.1 PSU. Sea ice was absent at S201, S209 and S311, while the concentration ranged between 52% (S215) and 98% (S235) at the remaining Arctic DCM sampling sites. Thus, most stations were located in areas of ongoing sea ice melt (Table 1).

Properties of the five UIW sampling station ranged in temperature between −1.59 (S255) and −1.8 °C (S360 and S384). Overall, the sampling depth resulted in a lower salinity compared to the DCM samples, ranging between 30.54 (S323) and 33.07 PSU (S360). Sea ice concentration was lowest at S323 with 60% and highest at 360 and 384 with 100%, respectively (Table 1). However, the late sampling of the S384 suggests a re-freezing scenario rather than a melting scenario.

**DNA extraction and sequencing**. DNA was isolated with the Power Soil DNA extraction kit (MOBIO) following the manufacturer's instructions, including a 2 × 20 s bead beating step to force the breakup of fungal cell walls and a second elution step to increase total DNA yield.

To study fungal diversity the ITS2 marker region was amplified with the primer combination ITS3Kyo2 (5′-GATGAAGAACGYAGYRAA-3′) and ITS4 (5′-TCCTCCGCTTATTGATATGC-3′)[54,55]. To investigate the branching position of phylogenetically disparate putative chytrid groups and to provide additional data on potential hosts organisms the wider eukaryotic community was sampled using PCR amplification of the SSU rDNA V4 region. The V4 region was amplified with the primer combination D512F (5′-ATTCCAGCTCCAATAGCG-3′) and D978R (5′-GACTACGATGGTATCTAATC-3′)[56]. PCR reactions were carried out in a 25 μl reaction volume in triplicate, including 12.5 μl NEBNext PCR mix (New England BioLabs), 5 μl of 1 μM primer each (forward and reverse) and 4 ng DNA template. The PCR amplification protocol was the same for both gene targets and consisted of an initial denaturation step at 95 °C for 3 min, followed by 32 cycles of denaturation at 95 °C for 30 s, annealing at 53 °C for 2 min and elongation at 72 °C for 30 s and a final elongation at 72 °C for 5 min. A total of 22 samples were subjected to Illumina sequencing (2 × 300 bp) using the Nano Flow Cell approach. All generated sequence data have been deposited in NCBI Sequence Read Archive (SRA) under the accession number PRJNA561496.

**Data processing, statistical analysis and phylogenetic tree calculation**. Raw sequence data were processed using the DADA2 package in R for sequence variant identification removing the requirement for one universal standard threshold for all cluster groups, i.e., 'taxa'[57]. In brief, sequence processing was carried out as follows: firstly, demultiplexed raw reads were subjected to DADA2 and quality profiles analysed, and secondly, reads were pre-processed by trimming, filtering and de-replication in order to identify unique sequences. Next, after this de-noising process, sequence variants in each sample were inferred and the paired-reads were merged. Finally, chimeras were identified and removed from the read data before taxonomic assignment. Settings for both data sets were the same (maxN = 0, truncQ = 2, maxEE = 2), except for the length parameter (minLen = 250 bp for the V4 and minLen = 50 bp for the ITS2 data set). Taxonomic classification of the V4 sequences was based on the SILVA reference database version 128[58]. Since reference sequences of the ITS2 region are not as comprehensive as for the V4 region, taxonomic classification was carried out by combining multiple public databases such as: UNITE, NCBI GenBank nt and ITS2 DB V.

All statistical analyses were performed in R (version 3.2.3; R Core Team, 2013) and the data are shown in Supplementary Data 1. To identify dissimilarities of microbial communities the hclust (hierarchical clustering) function of the package 'vegan' was used following ANOSIM to test for cluster significance[59]. Differences in V4 community composition in comparison to environmental factors were computed using the Bray Curtis and Euclidean distance, respectively and displayed in the form of NMDS plots. To test the significance of environmental factors MANTEL tests were calculated with 10,000 permutations and fitted onto the

NMDS plot in the 'ade4' R package[60]. The heatmap was created with the package 'heatmap3'[61]. To test for associations between diatom, chrysophyte, dinoflagellate and chytrid distributions within this study, an isometric log transformation was applied to the sequence counts using the 'robcompositions' package, with the 'corrplot' package being applied afterwards, including a significance test based on the Spearman's correlation coefficient and 1000 Monte Carlo permutations[62]. To test for correlations between the distribution of chytrids and protists within the Ocean Sampling Day initiative, sequences from surface water samples were downloaded from the Micro B3 project repository (https://github.com/MicroBr-IS/osd-analysis/wiki/Guide-to-OSD-2014-data) and processed with DADA2 using the PR² v4.10 reference database for taxonomic classification. Sequence variant correlations were then computed with SparCC, using the 'sparse correlations for compositional data' algorithm, 20 iterations and 1000 bootstrap replication for computing pseudo *p*-values[63].

Phylogenetic tree were calculated using a maximum likelihood (ML) approach. Evolutionary model evaluation was conducted using the automatic function in IQ-TREE[64] followed by tree calculation of the chytrid V4 sequences (using sequences with ≥0.01% relative representation) and chytrid ITS2 sequences. Topology calculation was combined with 1000 bootstrap (BS) replicates and a node-by-node SH-like approximate likelihood ratio tests. The sequence alignment was further supplemented with chytrid sequences from Richards et al.[17], Comeau et al.[14] and Kilias et al.[40] and with additional reference sequences identified using BLASTn from the GenBank nr database. Alignments are available with and without mask at 'figshare' (https://doi.org/10.6084/m9.figshare.11335658).

**Scanning electron microscopy**. To test the presence of putative diatom-infecting fungal-like pathogens in meltpond aggregates, we conducted a scanning electron microscopy (SEM) analysis of the material collected during the Nansen Legacy cruise (AeN2018707) in the summer of 2018. The sampled meltpond was located in the Nansen Basin at the station Sice2b at 83° 19′ 22″ N/29° 28′ 7″ E. The aggregates were collected from the bottom of the meltpond using a sampling tube, fixed immediately in a fixative mixture composed of glutaraldehyde (1% final concentration; Electron Microscopy Sciences, USA) and paraformaldehyde (4% final concentration; Electron Microscopy Sciences, USA) in HEPES buffer (60 mM final concentration) and stored at 4 °C. For the SEM preparation, aggregates were collected on a poly-ʟ-lysine-coated glass slide and post-fixed with a 100 μl volume of Osmium tetroxide (1% final concentration; Electron Microscopy Sciences, USA) in 0.3 M bicarbonate buffer placed on the glass slide for 30 min. After fixation, the glass slide was placed on a critical-point holder and washed twice in the 0.3 M bicarbonate buffer for 5 min. The subsequent dehydration through the ethanol series started with two 5 min washes in 50% ethanol and proceeded with single 5 min washes in 70, 80, 90 and 96% ethanol. The final steps were four 15 min washes in 100% ethanol. After dehydration, the sample was critical-point dried (BAL-TEC 030 CPD; Technion, Israel), mounted on an SEM stub, sputter coated with ca. 6 nm of palladium (Cressington 308R Desktop Advanced Coating Systems, USA) and observed under scanning electron microscope (Hitachi S-4800, Japan) at the Electron Microscopy Unit, Department of Biosciences, University of Oslo.

**Reporting summary**. Further information on research design is available in the Nature Research Reporting Summary linked to this article.

## Data availability

The raw data sets generated and/or analysed during the current study are available on the NCBI Sequence Read Archive (SRA) under the accession number PRJNA561496. Further, sequence alignments are deposited with and without mask at figshare[65]. The source data underlying plots shown in figures are provided in Supplementary Data 1.

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

## Acknowledgements

We would like to thank the captain and the crew of the *RV Polarstern* and all colleagues that participated on the ARK27-2 + 3 expedition. We are also thankful to the crew and the colleagues on board RV Kronprins Haakon who took part in the AeN2018707 cruise, especially to Ulrike Dietrich and Tobias Vonnahme from the University of Tromsø for collecting the meltpond samples. We are grateful for the support of Kerstin Oetjen for sampling support, David Milner for molecular biology, Adam Monier and Nick Irwin for bioinformatic support and Jens Wohlmann and Antje Hofgaard from the Electron microscopy Unit of the Department of Biosciences, University of Oslo for their advice on fixation and SEM sample preparation. This project has received funding from the European Union's Horizon 2020 research and innovation programme under the Marie Sklodowska-Curie Fellowship (No. 657141) awarded to E.S.K. and T.A.R. The meltpond SEM research was funded by the Research Council of Norway through the Nansen Legacy project (NFR-276730). Further funding was provided by InnovateUK grant TS/R00546X/1.

## Author contributions

E.S.K. conceived the study. K.M collected and provided samples for the study. L.J. and G.L. helped in performing statistical analysis and phylogenetic tree computation, respectively. L.S. performed Scanning Electron Microscopy. E.S.K. and T.A.R. wrote the paper with input from all authors. All authors approved the paper prior to submission.

## Competing interests

The authors declare no competing interests.
