## [Peer Review File · Communications Biology]

Reviewers' comments:

Reviewer #1 (Remarks to the Author):

The present study compared the community structures of fungi and eukaryotes among different stations in Arctic waters during sea ice minimum. Major findings could be 1) the community structures of fungi, especially the dominance of chytrids, were explained by ice cover and salinity, and 2) chytrids correlated with ice-associated diatoms. From those results, the authors concluded that chytrid-diatom associations may increase in the future as a consequence of ice retreat. I think their findings are interesting, and strengthen the previous findings about the dominance of chytrids in arctic ice (Comeau et al. 2016, Hassett& Gradinger 2016) and associations between parasitic chytrids and diatoms (Hassett& Gradinger 2016).

Major concerns

The authors analyzed eukaryotes community with V4 rDNA region. Fungi must have been detected as well, as the authors analyzed phylogenetic trees of chytrids based on SSU (L. 195-). I guess the authors selected certain taxa (Fig 1b) for SSU analysis. Please show the community composition of all species detected with V4 sequences first (maybe in supplementary info).

Then, the fungal community can be compared between V4 and ITS analysis, not only chytrids but also other phyla, especially unclassified fungi (L. 193-). The proportion of unclassified fungi might be different between ITS and SSU analysis, and "unclassified fungi" with ITS may turn to be assigned with SSU. It might be possible to detect other phyla, such as Rozellomycota (Cryptomycota), which has more database with SSU than with ITS.

Overall, Ascomycota dominated in many stations, but no discussions were made in the manuscript. As the author discussed Basidiomycota (L. 254-269), potential ecology and roles of Ascomycota should be discussed as well.

The proportion of each diatom reads among diatom (Bacillariophytes) were written in result (L. 131-). Please refer to Figure 5, which is helpful to understand the correlations with chytrids.

The picture of diatom-chytrid (Figure 6b) is a strong proof to say that detected chytrid must be the parasite of diatom (Nitzschia?). Please write which samples the picture was taken, and how materials were prepared in material and methods. Also, please write a few sentences in the result concerning the picture in the MS.

Concerning the correlation analysis of the data from the Ocean Sampling Day, three chytrids showed high correlations with many protists. It must be interesting to examine the phylogenetic position of the three (sv622, 1879, 1150). If those can be close in the novel clade, they are likely to be parasites of diatom, etc.

Minor comments

Figure 1 C) Please provide the explanation for the abbreviations in Table, such as D, T, S.

Figure 2 A) The color for DCM (beige) and UIW (grey) cannot be distinguished.

Reviewer #2 (Remarks to the Author):

Title: Fungal distribution and co-occurrence with diatoms in the Arctic Ocean is correlated with sea ice melt

Brief summary of the manuscript

This manuscript investigates the presence and diversity of marine chytrids from Arctic waters, their co-occurrence with abundant phytoplankton hosts, and the abiotic factors influencing both Arctic fungal and protist communities. The authors combined SSU and ITS surveys to describe microbial communities and how they changed in response to physicochemical factors. Some of the more interesting findings of this study include: 1) the role of sea ice and sea ice melt in shaping chytrid communities across the Arctic; 2) the association between diverse diatom hosts and putatively parasitic chytrids; and, 3) the high proportion of unclassifiable fungal sequence variants recovered from Arctic waters.

Overall impression of the work

There is growing interest in the role of marine fungi, due in large part to the recovery of chytrid and chytrid-like OTUs and sequence variants from environmental sequencing surveys of various marine habitats. The contributions of chytrids in freshwater food webs are better understood than in marine environments, but this study provides important insight into the potential significance of chytrids in Arctic food webs. The methods and analyses employed are appropriate to the questions being asked. This work is particularly timely, given the rapid warming of the Arctic. Overall, the manuscript is carefully and concisely written and the authors do not overstate their findings. My specific comments are often stylistic in nature or are designed to help improve clarity.

Specific comments, with recommendations for addressing each comment

Line 49: I think it's important to change the way you introduce and use the term "chytrid" in the paper. While "chytrid" has long been used to describe what we now know are multiple, paraphyletic phyla of zoosporic fungi, the organisms that you discuss throughout the paper appear to belong exclusively to the Chytridiomycota proper. For example, your ITS survey recovered no other phyla that could potentially be called "chytrids" (e.g., Neocallimastigomycota, Blastocladiomycota, Olpidium, or Crypto/Rozellomycota), and all of the individual sequence variants you discuss are either identified to the Chytridiomycota or place sister to/among the Chytridiomycota in your ML phylogeny. I think prefacing your results by describing the "paraphyletic class of fungi called chytrids which can form flagellated zoospores" suggests to the reader that you'll be investigating—and ultimately recovering—a diversity of zoosporic fungal lineages. Instead, it might be more appropriate to describe the zoosporic fungi, which are paraphyletic group of fungi that produce flagellated zoospores, and then indicate that you'll use "chytrids" as shorthand for the Chytridiomycota, specifically.

Line 121: change "nano-sized (20-2 μm)" to "nanoplankton (2-20 μm)"

Line 123: change "picosized" to "picoplankton"

Line 172: change "formally named Chytridiomycota group" to "part of the phylum Chytridiomycota"

Line 414: presumably you mean 4 ng of DNA template here

Figure 7: Add a negative sign to the units at the negative end of the scale

Reviewer #3 (Remarks to the Author):

This well designed study provides novel evidence for unique coldwater chytrid phylotypes and associations with sea ice algae and marine diatoms, demonstrating the role of salinity and melting sea ice in changing these arctic marine chytrid communities, as well as some affect of biogeography.

Bioinformatics and statistical analyses were appropriate and valid; the combined ITS2 and V4 SSU molecular data is appreciated. This paper will influence thinking in the field of microbial oceanography, through the important lens of climate change and sea ice retreat. Detailed methods and cautious interpretation of results are appreciated. Please clarify if you mean chytrid abundance or diversity increases as salinity decreases. I strongly recommend this paper for publication with minor revisions (see below).

Title: change to "Chytrid fungal distribution and co-occurrence..."

L36 Arctic ocean as hotspot of microbial biodiversity – citation needed for this.

L47 Define 'bona fide' marine fungi

L49 Chytrids -are they a class? Clarify.

L51 Is there evidence you can cite here to strengthen, eg your references 16 and 18, rather than saying "suggested to.."

L66 Is host density an abiotic factor?

L69 list some of these diatom genera here

L71 "we investigated chytrid distribution patterns' is better

L73 are you referring to microbial community when you say 'wider community structure"?

L79 could say 'fungal sequences' here, as focus is chytrids

L350 -unclear "environmental induced stress of host species" – clarify what is meant by this.

References 42 & 43: correct spellings are Cunliffe, Burgaud

Response to reviewer comments

Title: *'Chytrid fungal distribution and co-occurrence with diatoms in the Arctic Ocean is correlated with sea ice melt'*

Referee expertise:

Referee #1: Ecology of aquatic environments and microorganisms

Referee #2: Marine mycology

Referee #3: Marine fungi

Reviewers' comments:

Reviewer #1 (Remarks to the Author):

The present study compared the community structures of fungi and eukaryotes among different stations in Arctic waters during sea ice minimum. Major findings could be 1) the community structures of fungi, especially the dominance of chytrids, were explained by ice cover and salinity, and 2) chytrids correlated with ice-associated diatoms. From those results, the authors concluded that chytrid-diatom associations may increase in the future as a consequence of ice retreat. I think their findings are interesting, and strengthen the previous findings about the dominance of chytrids in arctic ice (Comeau et al. 2016, Hassett& Gradinger 2016) and associations between parasitic chytrids and diatoms (Hassett& Gradinger 2016).

Major concerns

- The authors analyzed eukaryotes community with V4 rDNA region. Fungi must have been detected as well, as the authors analyzed phylogenetic trees of chytrids based on SSU (L. 195-). I guess the authors selected certain taxa (Fig 1b) for SSU analysis. Please show the community composition of all species detected with V4 sequences first (maybe in supplementary info).

Response: This information is included in two supplemental figures one showing the wider V4-eukaryotic community composition detected, including fungi (Figure S1) and one showing a heatmap focusing exclusively on the fungal community (Figure S2). We are happy to include these data and we thank the reviewer for the suggestion that has improved our paper.

- Then, the fungal community can be compared between V4 and ITS analysis, not only chytrids but also other phyla, especially unclassified fungi (L. 193-). The proportion of unclassified fungi might be different between ITS and SSU analysis, and “unclassified fungi” with ITS may turn to be assigned with SSU. It might be possible to detect other phyla, such as Rozellomycota (Cryptomycota), which has more database with SSU than with ITS.

Response: A direct comparison of V4 and ITS2 read data is difficult since both sequence regions have different informative values in terms of the phylogenetic/taxonomic assignment and database representation and there are very few sequences of known taxa that are sequenced through V4-to-ITS2. This makes it difficult to connect between the two markers. As such they remain largely incomparable among uncharacterised fungal groups such as the ‘chytrids’ and ‘unclassified fungi,’ as it is impossible to refer from one sequence variant in the ITS2 data to the complementary sequence variant from the V4 data set. To try to correct for this problem, we have searched all publicly available chytrid genomes including Cryptomycota/Rozellomycota and Blastocladiomycota genomes and the TARA and NCBI metagenome databases for sequence contigs that would allow us to join V4 and ITS2 regions. In all cases we could not find sequence data that would allow us to connect these regions. To report this additional analysis, we have generated and included both an ITS2 and V4 phylogenies including sequences from genome projects relevant to this issue (Figure S3-S4). This additional work again confirms that the arctic associated chytrids in both ITS2 and V4 datasets cluster separately from known taxonomic groups. We have described these results in the paper and we have acknowledged that there is no direct evidence confirming that the ITS2 and V4 sequences we have recovered are from the corresponding lineages P8/L210-218 & P9/L256ff. However, both datasets confirm that there are patterns of chytrid biodiversity that are Chytridiomycota and associate both with freshening conditions and presence of pennate diatoms.

- Overall, Ascomycota dominated in many stations, but no discussions were made in the manuscript. As the author discussed Basidiomycota (L. 254-269), potential ecology and roles of Ascomycota should be discussed as well.

Response: the findings of the ascomycete analysis are now discussed in the paper P7/L199ff.

- The proportion of each diatom reads among diatom (Bacillariophytes) were written in result (L.131-). Please refer to Figure 5, which is helpful to understand the correlations with chytrids.

Response: A reference to the figure has been added. During the process of revision, the order of figures has changed. Figure 5 has been renamed as Figure 3. Please see P6/L152 & L164

- The picture of diatom-chytrid (Figure 6b) is a strong proof to say that detected chytrid must be the parasite of diatom (Nitzschia?). Please write which samples the picture was taken, and how materials were prepared in material and methods. Also, please write a few sentences in the result concerning the picture in the MS.

Response: We agree that this is good supporting evidence for our central hypothesis presented in our paper. The provenance of the SEM sample shown in Figure 6b has been added to the legend. We were inspired by reviewers’ comments here and therefore we have included some additional SEM work

focusing on melt pond aggregates (Figure 7). Methods and discussion of this work is also included (P12/L370ff & P19/L583ff). Using this SEM Data, we now present further evidence that putative chytrids are attaching to pennate diatoms in these ecosystems. This connects to another hypothesis in our paper that these trophic interactions are derived – in part – from aggregate formation derived from sea ice melt. P14/L441

- Concerning the correlation analysis of the data from the Ocean Sampling Day, three chytrids showed high correlations with many protists. It must be interesting to examine the phylogenetic position of the three (sv622, 1879, 1150). If those can be close in the novel clade, they are likely to be parasites of diatom, etc.

Response: The above-mentioned three chytrids were added to a new phylogenetic tree figure (Fig. S3) and a short statement has been included in the text describing their position. Please see P37/Figure S3. We note that these three represent divergent putative chytrids. Two of which cluster with other chytrids sampled from ‘cold water environments’. These lineages are not directly related to the main groups we identified during the arctic study, but provide further evidence of chytrid-like biodiversity in cold water habitats.

Minor comments

- Figure 1 C) Please provide the explanation for the abbreviations in Table, such as D, T, S. Figure 2 A) The color for DCM (beige) and UIW (grey) cannot be distinguished.

Response: The explanation of the utilised abbreviations has been added to the figure legend, while the colours have been intensified in figure 2 a) for better visualization. Please see P27/Figure1 and P28/Figure2, respectively.

Reviewer #2 (Remarks to the Author):

Title: Fungal distribution and co-occurrence with diatoms in the Arctic Ocean is correlated with sea ice melt

Brief summary of the manuscript

This manuscript investigates the presence and diversity of marine chytrids from Arctic waters, their co-occurrence with abundant phytoplankton hosts, and the abiotic factors influencing both Arctic fungal and protist communities. The authors combined SSU and ITS surveys to describe microbial communities and how they changed in response to physicochemical factors. Some of the more interesting findings of this study include: 1) the role of sea ice and sea ice melt in shaping chytrid

communities across the Arctic; 2) the association between diverse diatom hosts and putatively parasitic chytrids; and, 3) the high proportion of unclassifiable fungal sequence variants recovered from Arctic waters.

Overall impression of the work

There is growing interest in the role of marine fungi, due in large part to the recovery of chytrid and chytrid-like OTUs and sequence variants from environmental sequencing surveys of various marine habitats. The contributions of chytrids in freshwater food webs are better understood than in marine environments, but this study provides important insight into the potential significance of chytrids in Arctic food webs. The methods and analyses employed are appropriate to the questions being asked. This work is particularly timely, given the rapid warming of the Arctic. Overall, the manuscript is carefully and concisely written and the authors do not overstate their findings. My specific comments are often stylistic in nature or are designed to help improve clarity.

We thank the reviewer for the positive endorsement of our paper.

Specific comments, with recommendations for addressing each comment

Line 49: I think it's important to change the way you introduce and use the term "chytrid" in the paper. While "chytrid" has long been used to describe what we now know are multiple, paraphyletic phyla of zoosporic fungi, the organisms that you discuss throughout the paper appear to belong exclusively to the Chytridiomycota proper. For example, your ITS survey recovered no other phyla that could potentially be called "chytrids" (e.g., Neocallimastigomycota, Blastocladiomycota, Olpidium, or Crypto/Rozellomycota), and all of the individual sequence variants you discuss are either identified to the Chytridiomycota or place sister to/among the Chytridiomycota in your ML phylogeny. I think prefacing your results by describing the "paraphyletic class of fungi called chytrids which can form flagellated zoospores" suggests to the reader that you'll be investigating—and ultimately recovering—a diversity of zoosporic fungal lineages. Instead, it might be more appropriate to describe the zoosporic fungi, which are paraphyletic group of fungi that produce flagellated zoospores, and then indicate that you'll use "chytrids" as shorthand for the Chytridiomycota, specifically.

Response: We changed this section of the text following the reviewers suggestion. Please see: P3/L54-60

- *Line 121:* change "nano-sized (20-2 µm)" to "nanoplankton (2-20 µm)"

Response: 'Nano-sized' was replaced by 'nanoplankton'. Please see P5/L131

- *Line 123:* change "pico-sized" to "picoplankton"

Response: 'pico-sized' was replaced by 'picoplankton'. Please see (P5/L134)

- *Line 172:* change "formally named Chytridiomycota group" to "part of the phylum Chytridiomycota"

Response: Text was reformulated and "formally named Chytridiomycota group" was deleted along the course. Please see P8/L208ff

- *Line 414:* presumably you mean 4 ng of DNA template here

Response: The unit was corrected to 'ng'. Please see P17/L528

- *Figure 7:* Add a negative sign to the units at the negative end of the scale

Response: Figure 7 was reviewed by adding negative signs. Figure 7 was re-labelled as Figure 8. Please see Figure 8 P34

Reviewer #3 (Remarks to the Author):

This well designed study provides novel evidence for unique coldwater chytrid phylotypes and associations with sea ice algae and marine diatoms, demonstrating the role of salinity and melting sea ice in changing these arctic marine chytrid communities, as well as some affect of biogeography. Bioinformatics and statistical analyses were appropriate and valid; the combined ITS2 and V4 SSU molecular data is appreciated. This paper will influence thinking in the field of microbial oceanography, through the important lens of climate change and sea ice retreat. Detailed methods and cautious interpretation of results are appreciated...

...strongly recommend this paper for publication with minor revisions (see below).

We thank the reviewer for the positive endorsement of our paper.

- Please clarify if you mean chytrid abundance or diversity increases as salinity decreases.

Response: we are referring within this manuscript to chytrid representation which in the context of the V4 RDNA tag sequencing is equivalent to ‘relative abundance’. We have clarified this through-out e.g. see P8/L228, P15/L458.

- Title: change to “Chytrid fungal distribution and co-occurrence...”

Response: Title was changed to the reviewers suggestion Please see P1/L1

- L36 Arctic ocean as hotspot of microbial biodiversity – citation needed for this.

Response: Citation added to the comment (see: P3/L39)

-L47 Define ‘bona fide’ marine fungi

Response: This has been defined please see: P3/L49-51

- L49 Chytrids -are they a class? Clarify.

Response: Replaced ‘class’ with ‘group’ and included a description of the further usage of the term ‘chytrids’. Please see above P3/L53-60

- L51 Is there evidence you can cite here to strengthen, eg your references 16 and 18, rather than saying “suggested to..”

Response: We have included two references that show that chytrids act as parasites in aquatic environments and have been added to the statement. Please see P3/L57

- L66 Is host density an abiotic factor?

Response: Thank you for spotting this we have made a correction. Host density has now been described as biotic factor and added to the sentence. ‘*In-situ* experiments, addressing the effect of abiotic and biotic factors on susceptibility of diatom to chytrid infections have shown an influence of light-stress and host densities, factors which can potentially be driven by ice melt....’ Please see P4/L73

- L69 list some of these diatom genera here

Response: Representatives of the diatom genera have been included: “...many of the diatom genera (e.g. *Nitzschia*, *Pseudo-nitzschia*, *Chaetoceros*) shown to be susceptible to chytrid infections...”. Please see (P4/L77)

- L71 “we investigated chytrid distribution patterns’ is better

Response: Included the above-mentioned suggestion. Please see (P4/L79)

- L73 are you referring to microbial community when you say ‘wider community structure’?

Response: Changed ‘wider community structure’ to ‘wider protist community structure’. Please see (P4/L81)

- L79 could say 'fungal sequences' here, as focus is chytrids

Response: Included the above-mentioned advice and specified sequences as fungal. Please see (P4/L87)

- L350 -unclear "environmental induced stress of host species" – clarify what is meant by this.

Response: Sentence was modified to provide more clarity. Please see (P14/L412ff & L433)

- *References 42 & 43*: correct spellings are Cunliffe, Burgaud

Response: Spelling of the above-mentioned references corrected (now reference 44 and 45. Please see (P24).